# FluentSigners-50: A signer independent benchmark dataset for sign language processing

**Medet Mukushev**[1], **Aidyn Ubingazhibov**[2], **Aigerim Kydyrbekova**[1], **Alfarabi Imashev**[1], **Vadim Kimmelman**[3], **Anara Sandygulova**[1] *

**1** Department of Robotics and Mechatronics, School of Engineering and Digital Sciences, Nazarbayev University, Nur-Sultan, Kazakhstan, **2** Department of Computer Science, School of Engineering and Digital Sciences, Nazarbayev University, Nur-Sultan, Kazakhstan, **3** Department of Linguistic, Literary, and Aesthetic Studies, University of Bergen, Bergen, Norway

* anara.sandygulova@nu.edu.kz

**Data Availability Statement:** FluentSigners-50 files are available at the following DOI: https://doi.org/10.5281/zenodo.6043739 Due to the size constraint, we only share one repetition by each

## Abstract

This paper presents a new large-scale signer independent dataset for Kazakh-Russian Sign Language (KRSL) for the purposes of Sign Language Processing. We envision it to serve as a new benchmark dataset for performance evaluations of Continuous Sign Language Recognition (CSLR) and Translation (CSLT) tasks. The proposed FluentSigners-50 dataset consists of 173 sentences performed by 50 KRSL signers resulting in 43,250 video samples. Dataset contributors recorded videos in real-life settings on a wide variety of backgrounds using various devices such as smartphones and web cameras. Therefore, distance to the camera, camera angles and aspect ratio, video quality, and frame rates varied for each dataset contributor. Additionally, the proposed dataset contains a high degree of linguistic and inter-signer variability and thus is a better training set for recognizing a real-life sign language. FluentSigners-50 baseline is established using two state-of-the-art methods, Stochastic CSLR and TSPNet. To this end, we carefully prepared three benchmark train-test splits for models' evaluations in terms of: *signer independence*, *age independence*, and *unseen sentences*. FluentSigners-50 is publicly available at https://krslproject.github.io/FluentSigners-50/

## Introduction

Sign languages are natural languages used primarily by deaf communities around the world. Speech in sign languages is realized as sequences of gestures that include movements of the hands, body, arms, head, and face. Similar to spoken languages, sign languages have different levels of linguistic structure, including phonology, morphology, syntax, semantics, and pragmatics. Recently, there has been a substantial increase in interest in automatic sign language recognition [1].

Sign Language Processing (SLP) integrated three related research and development directions such as automated Sign Language recognition, generation and translation that aim to

contributor. A full version with five repetitions per person are located at the link: https://krslproject. github.io/fluentsigners-50/.

**Funding:** A.S. was awarded the funding by Nazarbayev University Faculty Development Competitive Research Grant Program 2019-2021 for the project "Kazakh Sign Language Automatic Recognition System (K-SLARS)". Award number is 110119FD4545. The funders had no role in study design, data collection and analysis, decision to publish, or preparation of the manuscript.

**Competing interests:** The authors have declared that no competing interests exist.

create technological solutions that would help break communication barriers for Deaf community and sign language users [2]. In order to advance these disciplines it is necessary to create large Sign Language corpora for data-driven approaches to learn from. As rightly pointed out by Bragg et al. [2], one of the main challenges of SLP is related to significant shortcomings of public sign language datasets that limit the power and generalizability of recognition systems trained on them. Apart from the apparent limitations of datasets such as the size of the vocabulary primarily due to expensive recording and annotation processes, most datasets contain only isolated signs (such as MS-ASL [3] and Devisign [4]), which are not appropriate for most real-world use cases that involve natural signing (continuous and spontaneous) and need to train on complete sentences and longer utterances [2].

As a result, Continuous Sign Language Recognition (CSLR) is a more complex problem than Isolated Sign Recognition (ISR), that is a recognition of glosses (i.e., a form of translation to represent signs [2]) per each frame/time-step in the video. CSLR is a sequence to sequence problem, where the input sequence is the video, i.e., sequence of images, and the output is a list of ordered glosses. Sign Language Translation (SLT), on the other hand, provides a list of ordered words that correspond to its translation in a spoken language (e.g., English). Most works on CSLR and SLT exploit a commonly used benchmark dataset—the RWTH-PHOE-NIX-Weather 2014 [5] that contains continuous signing performed by nine interpreters recorded in professional studio conditions.

Another limitation of most currently utilized datasets is in the lack of environmental variability as they are typically recorded in the same setting(s) and have one vocabulary domain, which results in overfitting when applied to models that are architecturally more complex [6]. As a result, many CSLR approaches focus on cropped hands only to reduce the problem [2]. It leads to losing information expressed by body movements, facial expressions, and mouthing, containing essential linguistic and grammatical information. Additionally, many sign language datasets have novice or non-native contributors (i.e., students) in addition to slower signing and simplifying the style and the vocabulary to make the computer vision problem easier, but of no real value [2].

This paper proposes a new large-scale Kazakh-Russian Sign Language (KRSL) dataset (FluentSigners-50) as a new CSLR benchmark. FluentSigners-50 proposes to address three shortcomings of commonly used datasets identified by Bragg et al. [2]: *continuous signing*, *signer variety*, and *native signers*. FluentSigners-50's main advantage is in its large signer variety: age (ranging from 8 to 57 years old), gender (18 male and 32 female), clothing, skin tone, body proportions, disability (deaf or hard of hearing), and fluency. Additionally, as the dataset was crowd-sourced: the participants were using a variety of their own recording devices (such as smartphones and web cameras), it resulted in a large variety of backgrounds, lighting conditions, camera quality, frame rates, camera aspect ratios, and angles. Finally, FluentSigners-50 contains recordings of 50 contributors that use sign language on a daily basis: either deaf, hard of hearing, hearing CODA (Child of Deaf Adults), and hearing SODA (Sibling of a Deaf Adult). As a result, the dataset contains a high degree of linguistic variability, including phonetic, phonological, lexical, and syntactic variations. It thus is a better training set for recognition of natural signing. While FluentSigners-50 directly contributes to SLP research related to KRSL, it can be utilized to test how well a model generalizes to unseen signers. Fig 1 demonstrates ten participants showcasing signer variety as well as video-related differences.

The main purpose of this dataset is to be used as a benchmark for sign language recognition/translation architectures. It can help researchers find out if their proposed models perform well and can generalize on unseen signers and different sign languages. Additionally, the dataset can be of interest to sign language linguistics as it has real-life, linguistic and inter-signer variability. Sentence types include statements, polar questions, wh-questions, and

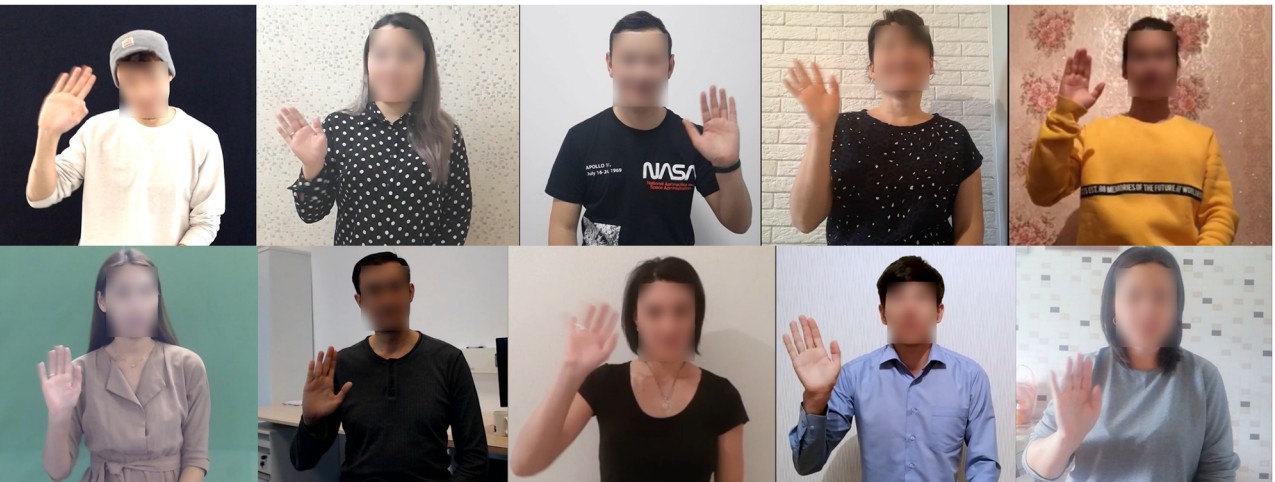

**Fig 1. Signers showing the sign HI.**

requests. This can allow linguists to analyze the data regarding its sentence type or non-manual features.

Additionally, this paper provides a baseline performance on two state-of-the-art architectures for the problems of CSLR and SLT, Stochastic CSLR (SCSLR) [7] and TSPNet [8] respectively. Stochastic CSLR is an end-to-end trainable state-of-the-art model that is based on the transformer encoder and Connectionist Temporal Classification (CTC) [9] decoder. It achieves a Word Error Rate (WER) of 25.3 on the RWTH-PHOENIX-Weather 2014 dataset [5] and WERs of 24.9 ± 6.2 on Split 1 (5-fold), 31.7 on Split 1 (one fold), 47.1 on Split 2, 52.0 ± 4.68 on Split 3 (5-fold), and 48.7 on Split 3 (one fold) of FluentSigners-50 dataset. TSPNet is a novel hierarchical video feature learning method obtained via a temporal semantic pyramid network. It achieves a 13.41 BLEU-4 score on the RWTH-PHOENIX-Weather 2014 dataset [5] and 16.0 ± 0.8 on Split 1 (5-fold), 15.7 on Split 1 (one fold), 10.5 on Split 2, 2.0 ± 1.1 on Split 3 (5-fold), and 2.2 on Split 3 (one fold) of FluentSigners-50 dataset.

The remainder of this paper discusses Related Work, followed by descriptions of data collection and the three splits. We then briefly introduce baseline methods used for the experiments detailed in the Experiments section. The paper concludes with Conclusions.

## Related work

This section discusses related work on sign language datasets and state of the art in Sign Language Processing.

### Sign language datasets

Sign language datasets are of great importance in order to advance the tasks of SLR and SLT. There exist several different types of datasets: 1) motion-capture data collected using sensors attached to various body locations (e.g., [10, 11]) 2) RGB-D data collected with the help of depth cameras (e.g., Kinects [12, 13]), and 3) RGB data that are more popular datasets due to their direct utility in real-life situations. Such datasets contain videos of either isolated signs or continuous signing. Table 1 presents an overview of the most commonly used sign language

**Table 1. Datasets used for continuous sign language recognition.** This list excludes datasets of isolated signs. *Deaf* column indicates if deaf signers contributed to the dataset. *In the wild* column indicates if recording settings varied. *No* means that the settings were the same for all samples.

| Datasets | Language | Signers | Deaf | Vocabulary | Samples | In the wild |
|---|---|---|---|---|---|---|
| The SIGNUM (2007) [14] | DGS | 25 | Yes | 780 | 780 | No |
| The RWTH-BOSTON-400 (2008) [15] | ASL | 4 | Yes | 483 | 843 | No |
| The RWTH-PHOENIX-Weather 2014T [16] | DGS | 9 | No | 2887 | 8257 | No |
| Video-Based CSL (2018) [17] | CSL | 50 | No | 178 | 25000 | No |
| The BSL-1K (2020) [18] | BSL | 40 | Yes | 1064 | - | No |
| The How2Sign (2020) [19] | ASL | 11 | Yes | 16000 | 35000 | No |
| **FluentSigners-50** | **KRSL** | **50** | **Yes** | **278** | **43250** | **Yes** |

datasets that are appropriate for the problem of CSLR with an inclusion of the proposed FluentSigners-50.

The high performance of deep learning methods for sign language recognition and translation tasks requires thousands of samples of data for training machine learning methods. Bragg et al. [2] highlight that only a few publicly available and large-scale sign language corpora exist. Furthermore, they specify the main concerns of existing datasets: a relatively small vocabulary size, absence of spontaneous (real-life) signing, novice signers and interpreters (e.g., students), and lack of signers' variety. Because of the importance of fluency and the naturalness of signing, we should distinguish between datasets containing contributors whose experience in sign language is unknown (e.g., learned a few gestures for the sake of dataset collection) and signers who use sign language as their first language. Many datasets record professional interpreters performing interpreting tasks. The nature of the task might lead them to use calque or loan translation, or the signed version of the spoken language instead of the natural sign language. Furthermore, many interpreters acquire sign language as adults, which also has important consequences to their sign language production. Additionally, datasets should differentiate between desired content and "real-life" signs (i.e., self-generated rather than prompted) [2] and datasets collected *in the wild* (i.e., varying recording settings and devices).

RWTH-Phoenix-Weather-2014 [5] is a German Sign Language (DGS) dataset used as a benchmark for most recent works in SLP. It features nine signers who performed sign language translations of the weather forecast on TV broadcasts. RWTH-Boston-400 [15] is one of the first CSLR benchmark datasets for American Sign Language (ASL). But it has only four signers present in the videos. In contrast, Video-Based CSL (Chinese Sign Language) [17] provides a large number of participants (n = 50) involved in collecting the dataset. At the same time, they are all recorded in the same recording settings, and most participants seem to be unfamiliar with sign language as they sign in slow and artificial ways without involving any facial expressions. SIGNUM [14] is a signer-independent CSLR dataset of DGS with all participants being fluent in DGS and are either deaf or hard-of-hearing. However, all videos were shot with a single RGB camera in a supervised condition with the same lighting and uniform black background. These concerns of existing datasets limit the accuracy and robustness of the models developed for SLR and their contribution to the challenges of real-world signing. More recent datasets aim to address most challenges of the previous datasets: BSL-1K [18] provides the largest number of annotated sign data while How2Sign [19] provides the largest vocabulary size. Similar to older datasets, they were either recorded in a controlled lab environment or extracted from the TV broadcast. From this perspective, FluentSigners-50 is the first sign language dataset that includes 1) a large signer variety recorded in various environmental conditions and 2) fluent sign language contributors (deaf, hard of hearing, CODA, or SODA).

Future SLR and SLT models can now be benchmarked on more than one dataset, which will help build more reliable recognition and translation systems.

## Sign language recognition

As stated previously, CSLR is a more complex problem than ISR as it deals with long temporal dependencies. Lately, alignment proposal optimizations were the focus of SLR methods. Additionally, deep neural networks, reinforcement learning, or recurrent neural networks were also widely used to advance the field. Evaluation of the works mentioned in this section were performed on the RWTH-PHOENIX-Weather 2014 [5] and RWTH-PHOENIX-Weather 2014T [16] datasets, which are used as a community benchmark [1, 2].

Zhang et al. [20] proposed an approach that applies encoder-decoder structure to reinforcement learning. It was on of the first models that deployed the Transformer [21] for sequence learning in CSLR. The Transformer's attention mechanism was extremely useful for distinguishing an effective sign language signal from a sequence of video clip features. Their method achieved comparable results with other methods and has a WER of 38.3%.

Temporal segmentation creates additional challenges for CSLR. To address this issue, Huang et al. [22] proposed the Hierarchical Attention Network with Latent Space (LS-HAN). This proposed framework eliminated the pre-processing of temporal segmentation and achieved an accuracy of 0.617. Advantage of this method is that it removed both the error-prone temporal segmentation in pre-processing and the sentences synthesis in the post-processing phases. Comparing to other algorithms that make use of convolutional neural network's feature learning capabilities as well as the iterative recurrent neural network's temporal sequence modeling capabilities, their method uses a similar approach but goes a step further by bridging the semantic gap with a latent space and then applying Hierarchical Attention Network to hypothesize semantic sentences.

Alternatively, Zhou et al. [23] proposed an I3D-TEM-CTC framework with iterative optimization for CSLR. In this work, they designed a dynamic pseudo label decoding approach that uses dynamic programming to identify an acceptable alignment path. In contrast to approaches that choose labels with the highest posterior probability from the entire lexicon, or utilize probability distributions directly as pseudo labels, their method filters away apparent incorrect labels and provides pseudo labels that follow the natural word order of sign language. By increasing the quality of pseudo labels, the system's final performance was improved and achieved a WER of 34.5%.

However, the most promising results were achieved by combining different modalities. For example, Koller et al. [24] presented an approach that achieved state-of-the-art results focusing on the sequential parallelism to learn a sign language, a mouth shape, and a hand shape classifier. They improved the WER to 26.0%. It clearly shows that a combination of manual and non-manual features such as the inclusion of a mouth shape could significantly enhance the performance of the recognition systems.

Stochastic CSLR [7] is an end-to-end trainable state-of-the-art model that is based on the transformer encoder and Connectionist Temporal Classification (CTC) [9] decoder. They represent each sign gloss with several states, with the number of states being a categorical random variable that follows a learning probability distribution, resulting in stochastic fine-grained labels for training the CTC decoder. In addition, they suggest a stochastic frame dropping mechanism and a gradient stopping approach to address the severe overfitting problem while training the transformer model with CTC loss. These two approaches also greatly minimize the training calculation, both in terms of time and space. It achieves a WER of 25.3 on the

RWTH-PHOENIX-Weather 2014 dataset [5] and outperforms Koller et al. [24] results by 0.7%.

## Sign language translation

The main difference between recognition and translation tasks is that SLT requires learning the sequence of words and their order. Camgoz et al. [16] introduced RWTH-PHOENIX-Weather 2014 T dataset with spoken language annotation as a benchmark for SLT. They used attention-based encoder-decoder models to extract gloss-level features from video frames and applied a sequence-to-sequence model to perform German Sign Language translation to German. Alternative approaches, such as work from Ko et al. [25] utilized human keypoints estimation [26] for SLT. As a result, they claim that extracting high-level features from sign language video with a sufficiently lower dimension is essential. They were successful in training a novel sign language translation system based on OpenPose human keypoints [26] and achieved 55.28% accuracy on the test set of KETI Sign Language Dataset. Orbay and Akarun [27] proposed a pre-processing step called tokenization by multi-task learning and showed that a model could achieve a higher translation scored without laborious gloss annotation.

Recently, transformer networks have been applied for SLT and showed encouraging results. For example, Camgoz et al. [28] applied multi-task transformers for both recognition and translation in an end-to-end manner achieving 21.32 black-4 score. This is accomplished by combining the recognition and translation challenges into a single unified architecture utilizing a Connectionist Temporal Classification (CTC) loss. This method eliminates the need for ground-truth timing information while simultaneously addressing two interconnected sequence-to-sequence learning tasks, resulting in considerable performance increase. They encode each frame individually using pre-trained spatial embeddings from Koller et al. [24], which are dependent on the gloss annotations.

Li et al. [8] introduced TSPNet, hierarchical sign video segment representation and achieved state-of-the-art results for video-to-text translation. TSPNet is a novel hierarchical video feature learning method obtained via a temporal semantic pyramid network. It achieves a 13.41 BLEU-4 score on the RWTH-PHOENIX-Weather 2014 dataset [16]. Although Camgoz et al. [16] obtained a higher BLEU-4 score on the RWTH-PHOENIX-Weather 2014T dataset, TSPNet is state-of-the-art method for video-to-text task. Camgoz et al. [16] uses an intermediate step of segmenting video into glosses and then translating them into text, while TSPNet proposes novel sign video segment representation alleviating the need for accurate segmentation into glosses. We believe that this approach has a more applicable potential as not every dataset has gloss-based annotations. And despite its current lower performances compared to Camgoz et al. [16], TSPNet can accommodate more datasets that have only spoken language translations of sign language videos.

## FluentSigners-50 dataset

Given the importance of signer independence and signer variety, we involved the local Deaf community in FluentSigners-50 data collection.

### The data

The FluentSigners-50 dataset consists of everyday conversational phrases and sentences in KRSL, the sign language used in the Republic of Kazakhstan. KRSL is closely related to Russian Sign Language (RSL) and some other sign languages of the ex-Soviet Union. While no official research comparing KRSL with RSL exists, our observations based on our experience researching both languages are that they show a substantial lexical overlap and are entirely mutually

intelligible [29]. The issue of dialectal variation of KRSL in Kazakhstan has not been studied at all yet. (See Kimmelman et al. [30] for a discussion of lexical variation in RSL in Russia.) As we discuss below, the signers in the dataset come from different regions of Kazakhstan, so some regional variation is possibly represented in the dataset. However, it was not created for the purposes of studying such variation.

The sentences and phrases of FluentSigners-50 represent the following sentence types: statements, polar questions, wh-questions, and requests.

## Dataset collection

At first, we invited six professional sign language interpreters who work at the national television. They were born and grew up in families with at least one deaf parent (i.e. hearing CODA). We worked together to compose the phrases and sentences for the dataset commonly used in the Deaf community. These six interpreters translated the sentences into KRSL and recorded five repetitions per sentence using the Logitech C920 Pro web camera. In addition, we asked one of them to record the instruction video with the welcoming address, explanation of the task, and one exemplary performance of the 173 sentences in KRSL. We later distributed this video to all other contributors of the dataset. Thus, for all participants except for the CODA interpreters, the task was to repeat the KRSL sentences they saw in the recording. It was a tradeoff that we had to make in order for the dataset to have the same KRSL sentences. Although this approach does not result in constructions that would typically be produced by native signers, we had to rely on the interpreters' opinions and translations. In addition, the contributors were allowed to make changes to the original interpretation that often included changing the order of signs, omitting or replacing signs, adding extra signs as well as varying the way the signs were performed. As a result of this process, the dataset contains a high degree of linguistic variability. The summary of FluentSigners-50 dataset is presented in Table 2.

The other contributors of the dataset were friends and relatives of the six interpreters who participated in this dataset voluntarily and signed an informed consent form approved by the Ethical committee of Nazarbayev University. All contributors received monetary compensation for the participation and agreed to have their data shared as a dataset. Note that the informed consent form was translated to KRSL to enable full accessibility. The individuals in this manuscript have given written informed consent (as outlined in PLOS consent form) to publish their photographs (Fig 1).

All FluentSigners-50 contributors use sign language on the daily basis. Not all of them can be defined as native signers, that is, as signers who have acquired KRSL from birth from their parents, but note that this notion in general is complicated and questionable [31]. Instead of

**Table 2. Statistics of the FluentSigners-50 dataset.**

| Video resolution | Range |
|---|---|
| Number of Signers | 50 |
| Repetitions | 5 |
| Number of sentences | 173 |
| Video duration (seconds) | 2∼11 |
| Body joints | Upper-body involved |
| Mean number of signs per sentence/phrase | 4 |
| Vocabulary size | 278 |
| Total number of videos | 43250 |
| Total number of hours | 43.9 (∼150 raw) |

**Table 3. Survey results with KRSL status for participants of FluentSigners-50 dataset.**

| 1 | To the best of your memory, or from what your parents have told you, which of the following best describes your use of sign language in your home during your early childhood? | |
|---|---|---|
| | We only signed and used no spoken language. | 30 (60%) |
| | We mostly signed, but we used some spoken language as well. | 11 (22%) |
| | We signed and spoke in roughly equal amounts. | 4 (8%) |
| | We mostly spoke, but used some sign language too. | 3 (6%) |
| | We only spoke and used no sign language. | 0 (0%) |
| | We rarely spoke or signed, but relied on gestures to communicate | 2 (4%) |
| 2 | Do you use sign language on a daily basis? | |
| | Yes | 49 (98%) |
| | No | 1 (2%) |
| 3 | When did you learn sign language? | |
| | from birth | 34 (68%) |
| | In kindergarten | 4 (8%) |
| | In school | 9 (18%) |
| | In adulthood | 3 (6%) |

trying to divide them into native or non-native, we collected data on their hearing status, daily use of KRSL, where they acquired KRSL, and the preferred language of their family when they were growing up (adapted from Allen (2015) [32]). The results of this survey can be seen in Table 3. Concerning the hearing status, the participants are deaf (N = 32), hard of hearing (N = 6), hearing SODA (N = 3) or hearing CODA (N = 9). In total, 30 participants (deaf and hearing) had at least one signing deaf parent, and 34 of the participants have acquired KRSL from birth, 4 in kindergarten, 9 at school, and 3 in adulthood. They all came from various regions of Kazakhstan and are of different age and gender groups. Therefore, the dataset represents a diverse population of signers, mostly deaf and some hearing, with a majority of signers who have acquired KRSL early, but also with some signers who acquired it later. Fig 2 shows

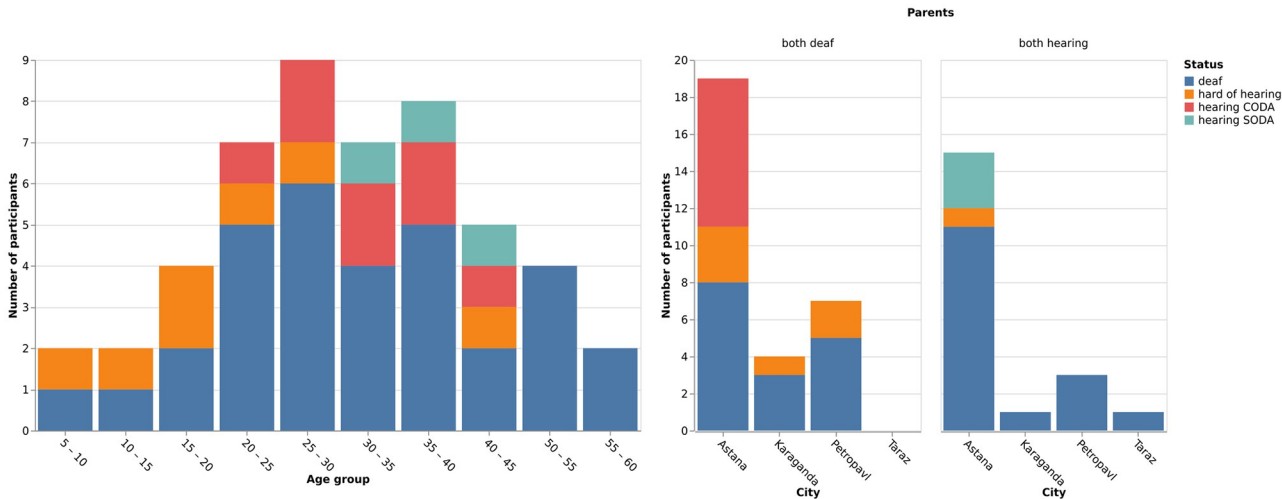

**Fig 2. Distribution of FluentSigners-50 contributors' demographics such as city, age, parents and status (deaf, hard of hearing, hearing SODA or CODA).**

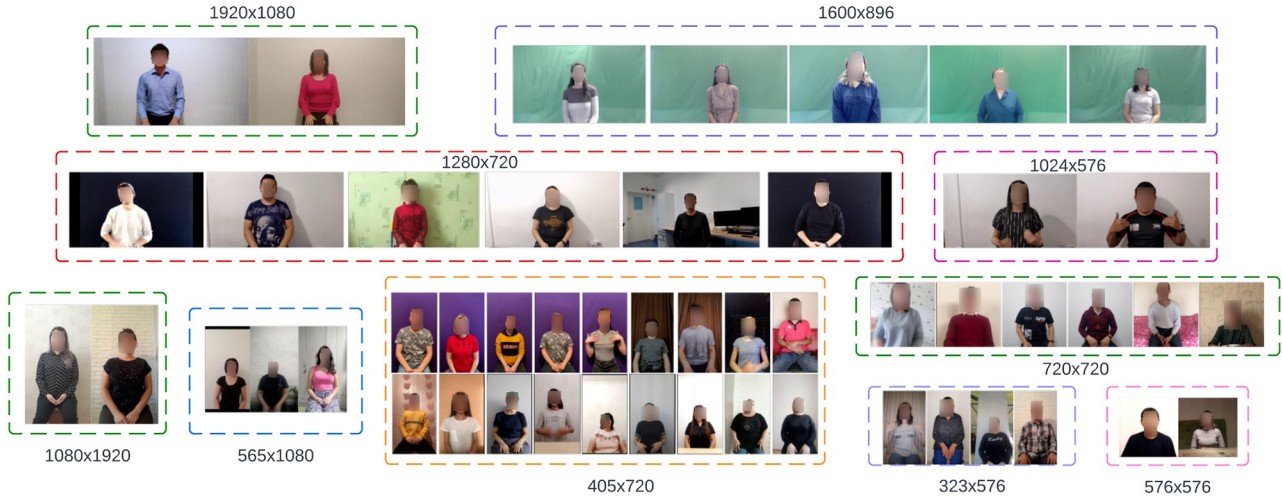

**Fig 3. Diversity of video resolutions, camera angles, lighting conditions and backgrounds present in FluentSigners-50.**

the demographics of participants. This dataset is thus more representative of the diversity of the whole signing population than most other comparable datasets for other sign languages.

The participants were asked to watch the pre-recorded sentences one by one and record themselves repeating each sentence five times. Such a data collection process did not require the presence of a researcher or an interpreter. Even though the signers were asked to repeat the pre-recorded KRSL sentences, many of them added their minor corrections. They performed the sentences in their way since they relied on their own communication experience, method of interpretation, etc. All collected videos have different quality and resolutions since they used their mobile phones or web-cameras with varying backgrounds, illumination conditions, camera angles, making the FluentSigners-50 dataset diverse and realistic compared to other CSLR datasets. The filming process of each contributor took about 3.5 hours. The duration of all raw videos is more than 150 hours. Each video was carefully validated and annotated, resulting in a total of 43 hours of labeled trimmed materials. Fig 3 shows one frame from videos of each participants.

## Linguistic properties of the data

In addition to the real-life variability in recording conditions, the data set also contains a high degree of linguistic variability. It thus is a better training set for recognition of real-life sign language.

The phrases recorded mainly include everyday conversational phrases, such as greetings, simple questions, and answers that are part of daily conversations related to age, living, family, food, weather, work, and other categories.

The phrases represent the following sentence types: statements, polar questions, wh-questions, and requests. In addition, some phrases are single-sign utterances, such as "Hello!" and "Goodbye!". Some of the statements and requests express negative polarity (contain negation). Many sentences contain 1st or 2nd person pronouns, but no 3rd person pronouns appear in the data set.

While a complete linguistic analysis of the data set is yet to be conducted, we can already observe a large amount of variation at different levels, as would also be expected in naturalistic sign language production [33].

Phonetic and phonological variations are observed in many signs. For example, the sign HELLO is produced with 2, 3, or 4 movement repetitions by different signers, which is most likely phonetic variation. The sign TWO.OF.US is produced either with the thumb-and-index handshape or with the index-and-middle handshape (phonological variation). An interesting case is the sign DEAF, which can be produced with movement either from the ear to the mouth or in the opposite direction; in addition, the ear location can be lowered to the cheek; finally, the index finger might touch the initial and final locations, or only approach them (see [34] for similar variation in American Sign Language).

Lexical variation is also found in the data set, where different lexical signs for the same concept are chosen by different signers. For example, in sentence 109, different lexical signs for 'adore' are used by different signers. On the boundary between lexical and grammatical, variation can be observed in the first person singular pronoun 'I,' which occurs either as pointing to oneself with the index finger, or as touching the chest with a flat hand, or with the handshape representing the Russian letter 'Я'. This letter is used as the first person pronoun in Russian, so the third variant of the pronoun is an instance of borrowing.

Syntactic variation can be observed as well. Word order varies between signers: one pattern concerns the position of wh-signs; for instance, in sentence 65 'Where were you born?', the wh-sign WHERE occurs either in the initial or medial position, as also described for some other sign languages [35]. In addition, in some sentences, some signers produce more signs than others, e.g., producing or omitting the past tense marker.

Finally, manual and non-manual prosody also vary in the data set. Some signers produce phrases much more fluently and fluidly, while others make more considerable pauses between signs. Concerning non-manuals, eyebrow raise associated with polar question marking is much more pronounced in some signers than in others. Negative facial expression and head-shake also vary in degree, but also in the scope: in sentence 126, 'No, I do not eat fish' some signers only accompany the negative signs NO and NOT with the non-manuals, while others also mark the subject and the verb; similar variability has been reported for other sign languages [36, 37].

## Suggested splits

In contrast to random train-test splits, we propose to set baseline performance for our dataset using three splits: signer-independent, age-independent, and unseen vocabulary splits. The distributions of each split are represented in Fig 4. We suggest conducting a 5-fold cross-validation on Split 1 and Split 3 to report mean and standard deviation results. Split 2 has only one

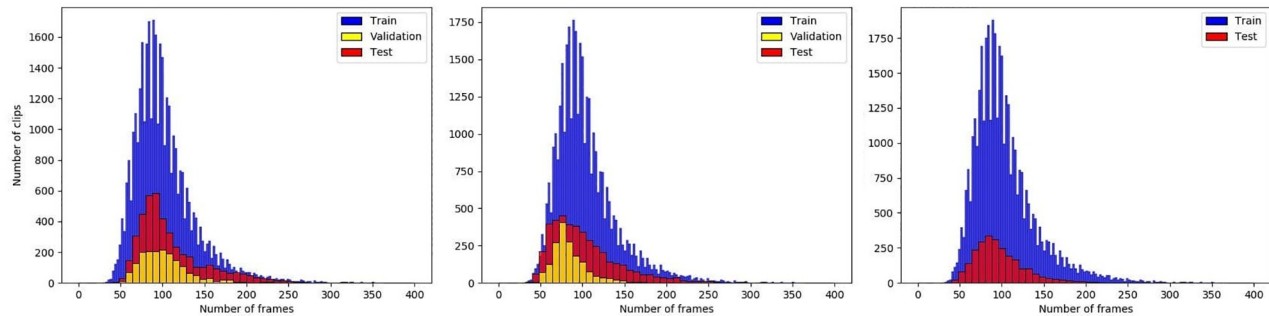

**Fig 4. Distribution of the number of frames over sentence-level clips in training, validation and test sets for each split: Split 1 (left), Split 2 (middle), Split 3 (right).**

fold, as it is not suitable for creating 5-folds due to not having enough data points to satisfy split's definition (age independence).

**Split I: Signer independence.** Signer independence is one of the main challenges that must be addressed for the real-life value of SLR models. Our dataset can help address this challenge as it provides both visual-related differences of each signer, such as variability in postures, distance to the camera, lighting, backgrounds, camera aspect ratios and angles, frame rates, quality, as well as linguistic-related differences of each signer, such as phonetic, phonological, lexical and syntactic variations. In order for implemented architectures to generalize well for unseen signers, training, validation, and testing splits should include different signers.

It is important for the proposed architectures to perform well on signers that are not seen during the model training. For this reason, we take all sentences of 40 signers as a training set, and the remaining 10 signers are divided into validation and testing sets (5 signers in the test set and 5 signers in the validation set). We also ensure that validation and testing sets have both CODA and not CODA signers. It is needed to test a model against the fluency of the participants in signing. As discussed above, the dataset contains variation in fluency between different signers. Also, some signs are performed differently by various participants, which brings additional challenges. To this end, the training set has 34600 samples, the validation set has 4325 samples, and the testing set has 4325 samples.

**Split II: Age independence.** An additional advantage of our dataset is the presence of child contributors, as we noticed that they increase the difficulty and variability of the dataset due to the differences in the fluency of signing between child and adult signers. The exact proportions and strategy are applied for the second split. However, in this case, we exclude child signers from the training set and explicitly create an age-independent split with signers aged 9 to 18 years old only included in validation and testing sets. Furthermore, in this case, the training set contains videos recorded on web cameras or mobile phone cameras. In contrast, the testing set contains only videos recorded on mobile phone cameras (as all children used their mobile phones). This brings additional complexity for model testing, as videos are from a diverse sample space. Similar to Split 1, the training set has 37195 samples, the validation set has 1730 samples, and the testing set has 4325 samples.

**Split III: Unseen sentences and signer independence.** As previously discussed, sign languages can have several properties of linguistic variability. FluentSigners-50 is rich in linguistic variability as contributors often refused to follow the exemplary video and interpreted the sentences in the way they are used to. Apart from that, FluentSigners-50's sentences were composed in a way that there are multiple instances of the same glosses used in several sentences surrounded by different glosses and in a different order (e.g., I love coffee (S112). I don't drink coffee, but I like tea (S113). Let's have some tea (S117).)

Such split can be used to see how well a proposed architecture can recognize and translate glosses if used in other contexts and gloss order. As the overall vocabulary is limited, we decided to have only training and testing sets. We select ten sentences as the testing set and the rest 163 sentences as the training set. To this end, the testing set includes signs used in unseen sentences. Each sign varies with respect to the order in a sentence it appears in and the context of the sentence. Additionally, a test set does not include the same signers as in the training set.

## Baseline method

This section focuses on the model architectures and reasoning behind decisions for the model design.

## SLR baseline: Stochastic CSLR

Stochastic CSLR [7] is an end-to-end trainable state-of-the-art model that is based on the transformer encoder [21] and CTC decoder. There were three stochastic components proposed within this architecture: stochastic frame dropping mechanism, stochastic gradient stopping method, and stochastic fine-grained labeling method.

**Stochastic frame dropping (SFD)** mechanism discards a fixed proportion of frames in a video uniformly without replacement. Not only does the SFD mechanism alleviate overfitting, but it also reduces the memory footprint and training time.

**Stochastic gradient stopping (SGS)** method stops gradient back-propagation during visual feature extraction for a fixed proportion of frames, which is sampled uniformly without replacement. This method is also a measure against overfitting and, at the same time, reduces memory footprint and training time.

**Stochastic fine-grained labeling (SFL)** assigns multiple states to each gloss, the number of which is a categorical random variable. Intuitively, each gloss constitutes multiple frames, and multiple state modeling helps discriminative feature learning of the network. SFL method allows the network to learn the number of states for each gloss. A different number of states for each gloss are sampled to learn the probability of the state number sequences that produce lower CTC loss.

ResNet18 [38] model pretrained on ImageNet [39] was used as the visual model for Stochastic CSLR.

## SLT baseline: TSPNet

TSPNet [8] architecture is a state-of-the-art model for Video-to-Text Sign Language Translation task. TSPNet introduces inter-scale attention to evaluate and enhance local semantic consistency of sign segments and intra-scale attention to resolve semantic ambiguity using non-local video context. It employs an encoder-decoder architecture. TSPNet has two main components:

**Multi-scale Segment Representation** helps to alleviate the influence of imprecise sign video segmentation by employing a sliding window approach to create video segments with multiple window widths. To eliminate substantial ambiguity in gesture semantics, they develop a hierarchical feature learning method that utilizes local temporal structure to enforce semantic consistency and non-local video context.

**Hierarchical Video Feature Learning** is used by the encoder to learn discriminative sign video representations by exploiting the semantic hierarchical structure among video segments. The output of the encoder is fed to a Transformer decoder to acquire the translation. They develop a segment representation for sign videos and learn both spatial and temporal semantics of sign gestures.

# Results and discussion

## Metrics

The following two metrics are commonly used to evaluate models' performances in SLT and SLR tasks:

**Word Error Rate (WER)** [5] metric is reported for SLR baseline.

$$WER = \frac{S + D + I}{N}$$

with minimum $S + D + I$ where $S$ is the number of substitutions, $D$ is the number of deletions,

*I* is the number of insertions required to convert the prediction into the reference, and *N* is the number of words in the reference phrase.

**Bilingual Evaluation Understudy (BLEU)** [40] metric is reported for SLT baseline. *n*-grams are *n* words appearing together sequentially. *n*-grams precision score $p_n$ is calculated as follows:

$$p_n = \frac{\sum\limits_{C \in \{Candidates\}} \sum\limits_{n-gram \in C} Count_{clip}(n-gram)}{\sum\limits_{C' \in \{Candidates\}} \sum\limits_{n-gram' \in C'} Count(n-gram')},$$

where $Count_{clip}$ is the maximum number of occurrences in any of the reference phrases of a given *n*-gram in the candidate (predicted) phrase. *Count* is the number of occurrences in a candidate phrase of a given n-gram in the predicted phrase. To penalize short candidates, Brevity Penalty is used:

$$BP = \begin{cases} 1, & \text{if } c > r \\ e^{1-\frac{r}{c}}, & \text{if } c \leq r \end{cases}.$$

Final n-gram BLEU score is computed as follows:

$$BLEU - n = BP \cdot exp\left(\sum_{i=1}^{N} w_i log p_i\right).$$

We consider $n \in \{1, 2, 3, 4\}$ with equal weights $w_i = \frac{1}{n}$ for our baseline.

## Experiments

**Training details.** Both Stochastic CSLR [7] and TSPNet [8] are implemented in PyTorch [41] framework. Tesla V100 GPU was used to train the models.

For Stochastic CSLR, the video frames are resized to $256 \times 256$, and random crops of size $224 \times 224$ are extracted. The model is trained using Adam optimizer with a batch size of 8 for 30 epochs. The learning rate is scheduled according to $\eta_i = \eta_0 \cdot 0.95^{\frac{i}{2}}$ for $i^{th}$ epoch with $\eta_0 = 1 \times 10^{-4}$. 50% of the frames are dropped randomly from each video and the gradient stopping parameter is 75%.

TSPNet is developed using FAIRSEQ [42] framework in PyTorch [41]. Video features are extracted with the I3D network [43] pre-trained on the Kinetics dataset. The model is trained with Adam optimizer with an initial learning rate of $10^{-4}$. The network trains for a maximum of 200 epochs, and the learning rate reduce factor is 0.5, and the patience is 8 epochs. Label smoothing [44] with a weight factor of 0.1, and weight decay parameter of $10^{-4}$ are applied to regularize the model.

**SLR results.** Table 4 shows the SLR task results of Stochastic CSLR [7] on FluentSigners-50 and RWTH-PHOENIX-Weather 2014T [16] in terms of WER score with a lower number being a better result. For the RWTH-PHOENIX-Weather 2014T [16] dataset we printed the numbers from the Stochastic CSLR [7] paper directly.

We firstly report the results obtained from 5-fold cross-validation in order to avoid a possibly biased test set. And since our dataset is relatively large with each sample being a video, computing 5-fold cross-validation is computationally intensive, we also present the results from 1-fold similarly to RWTH-PHOENIX-Weather 2014T [16] and Chinese Sign Language (CSL) [17] datasets that have one fixed test set that is shared with the community.

**Table 4. SLR results of Stochastic CSLR [7] on RWTH-PHOENIX-Weather 2014T [16] and different splits of FluentSigners-50.**

| Dataset | val (WER) | test (WER) |
|---|---|---|
| FluentSigners-50: Split 1 | 25.4 ± 2.8 | 24.9 ± 6.2 |
| FluentSigners-50: Split 1 (one fold) | 21.8 | 31.7 |
| FluentSigners-50: Split 2 | 10.6 | 47.1 |
| FluentSigners-50: Split 3 | – | 52.0 ± 4.68 |
| FluentSigners-50: Split 3 (one fold) | – | 48.7 |
| RWTH-PHOENIX-Weather 2014T | 25.1 | 26.1 |

The results on Split 2 of FluentSigners-50 demonstrate model generalization by age. The reported results of the testing set suggest that this split was much more difficult than Split 1. It could be explained by the difficulty of generalizing on child signers when the model is trained and validated on adult signers. We noticed that many children were less confident on camera, and it might have caused the lower performance of the model compared to Split 1. An interesting observation is that Split 2's WER is 10.6 on the validation set where the training set does not include child signers, while Split 1's WER is 26.9, where child signers add noise to the training process. Therefore, Split 2's WER is 47.1 compared to only 27.3 WER of Split 1.

The results on Split 3 of FluentSigners-50 show the performance of model generalization on unseen sentences. The reported results prove that Split 3 is much more challenging than Split 1 due to a relatively small number of sentences (N = 173). Therefore, each sentence in the testing set consists of glosses that the model encountered only several times during training in each epoch but used in a different sentence of the testing set. Table 5 presents several examples of how the model failed in its predictions. For example, in some cases (S005, S021, S041), the model confused signs with the wrong ones, while in others the model skipped some signs (S081, S134, S159).

**SLT results.** Table 6 shows the SLT task results of TSPNet [8] on FluentSigners-50 and RWTH-PHOENIX-Weather 2014T [16] across four BLEU scores, with the higher numbers being the better score. For the RWTH-PHOENIX-Weather 2014T [16] dataset we printed the numbers from the TSPNet [8] paper directly.

As BLEU-4 is the most challenging score, which is also commonly used for comparison between models, we can see that BLEU-4 results indicate that Split 1 is the easiest split and has 16.0 compared to 10.5 and 2.0 for Split 2 and 3, respectively. It is consistent with Stochastic CSLR results on the SLR task.

**Discussion.** Split 1 of FluentSigners-50 dataset results demonstrates how a model generalizes to unseen signers and variability of both camera and environmental settings. In contrast, the RWTH-PHOENIX-Weather 2014T has only 9 signers recorded in professional studio

**Table 5. Ground-truth (GT) and predictions in Split 3 of FluentSigners-50 for SLR.**

| Sentence ID | Ground-truth | Prediction |
|---|---|---|
| S005 | 'у меня все хоРошо' | 'у меня все *ПЛОХО*' |
| S021 | 'у меня есть новости' | 'у меня *ДЛЯ ТЕБЯ* е оо' |
| S041 | 'ЖЕЛАЮ хорошо день' | '*КАК ПРОШЕЛ* е' |
| S081 | 'у меня *СЕГОДНЯ день рождения*' | 'у меня день рождения' |
| S134 | 'ты мне *НЕ* ра' | 'ты мне нравится' |
| S159 | 'ты глухой' | '*У ТЕБЯ ЕСТЬ ТЕЛЕФОН*' |
| S159 | 'ты глухой' | '*У ВАМ* глухой' |

**Table 6. SLT results of TSPNet [8] on RWTH-PHOENIX-Weather 2014T [16] and different splits of FluentSigners-50.**

| Dataset | BLEU-1 | BLEU-2 | BLEU-3 | BLEU-4 |
|---|---|---|---|---|
| FluentSigners-50: Split 1 | 20.3 ± 1.0 | 17.8 ± 0.9 | 16.6 ± 0.8 | 16.0 ± 0.8 |
| FluentSigners-50: Split 1 (one fold) | 20.7 | 18.0 | 16.7 | 15.7 |
| FluentSigners-50: Split 2 | 14.2 | 12.0 | 11.0 | 10.5 |
| FluentSigners-50: Split 3 | 5.1 ± 0.45 | 3.9 ± 0.53 | 3.1 ± 0.78 | 2.0 ± 1.1 |
| FluentSigners-50: Split 3 (one fold) | 5.1 | 4.1 | 3.0 | 2.2 |
| RWTH-PHOENIX-Weather 2014T | 36.1 | 23.1 | 16.9 | 13.4 |

conditions leading to a lack of diversity in signer variability and environmental variability. Thus, any implemented models might overfit on these attributes and perform poorly on unseen signers recorded in various conditions. Therefore, FluentSigners-50 aims to address the potential limitations of RWTH-PHOENIX-Weather 2014T by providing an opportunity to learn from large signer variety and in the wild recording conditions. This will allow for more robust and applicable solutions. We encourage researchers to conduct performance tests of their proposed models and report their results on both RWTH-PHOENIX-Weather 2014T and Split 1 of FluentSigners-50 to account for this problem.

Similarly to SLR results, the results of TSPNet on FluentSigners-50 demonstrate similar behavior of splits' performances with Split 1 being the easiest split, followed by Split 2, and Split 3 being the most challenging split.

## Conclusion

This paper presents the FluentSigners-50 dataset, a new large-scale Kazakh-Russian Sign Language dataset that aims to contribute to the development of continuous sign language recognition by introducing a new large-scale multi-signer benchmark. FluentSigners-50 consists of 173 sentences performed by 50 contributors with more than 43 hours of video (43,250 samples). The main difference with other sign language datasets is its large number of sign language contributors who are deaf, hard of hearing, hearing CODA or SODA. Every video was recorded in different settings with varying backgrounds and lighting using their web or mobile phone cameras, which resulted in considerable variability in the resolution and frame rate of videos. Additionally, the FluentSigners-50 dataset contains a high degree of linguistic and inter-signer variability and thus is a better training set for recognition of a real-life sign language. We establish benchmark performances using state-of-the-art models (SCSLR and TSPNet) on three splits: signer-independent, age-independent, and unseen sentences + signer independence. The dataset is fully open and is available online at https://krslproject.github.io/FluentSigners-50.

## Supporting information

**S1 Table. This table contains the sentences used in this study.**
(PDF)

## Acknowledgments

We would like to thank the participants who kindly agreed to contribute to the dataset and have their recordings shared publicly.

## Author Contributions

**Conceptualization:** Medet Mukushev, Alfarabi Imashev, Vadim Kimmelman, Anara Sandygulova.

**Data curation:** Medet Mukushev, Aigerim Kydyrbekova, Alfarabi Imashev.

**Formal analysis:** Medet Mukushev, Aidyn Ubingazhibov, Vadim Kimmelman.

**Funding acquisition:** Anara Sandygulova.

**Investigation:** Medet Mukushev, Aidyn Ubingazhibov, Aigerim Kydyrbekova, Alfarabi Imashev.

**Methodology:** Medet Mukushev, Vadim Kimmelman.

**Project administration:** Medet Mukushev, Aigerim Kydyrbekova.

**Resources:** Medet Mukushev, Aigerim Kydyrbekova, Alfarabi Imashev.

**Software:** Medet Mukushev, Aidyn Ubingazhibov.

**Supervision:** Vadim Kimmelman, Anara Sandygulova.

**Validation:** Medet Mukushev, Alfarabi Imashev, Vadim Kimmelman.

**Visualization:** Medet Mukushev, Aidyn Ubingazhibov.

**Writing – original draft:** Medet Mukushev, Aidyn Ubingazhibov, Anara Sandygulova.

**Writing – review & editing:** Vadim Kimmelman, Anara Sandygulova.

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
