## [Decision Letter · Decision Letter 0]

7 Dec 2021

PONE-D-21-21778NativeSigners-50: a signer independent benchmark dataset for Sign Language ProcessingPLOS ONE

Dear Dr. Sandygulova,

Thank you for submitting your manuscript to PLOS ONE. After careful consideration, we feel that it has merit but does not fully meet PLOS ONE’s publication criteria as it currently stands. Therefore, we invite you to submit a revised version of the manuscript that addresses the points raised during the review process.

We look forward to receiving your revised manuscript.

Kind regards,

Aaron Jon Newman

Academic Editor

PLOS ONE

2. We note that Figure 1 includes an image of a [patients / participants / in the study].

“We would like to thank the dataset contributors for agreeing to participate in data 451

collection. This work was supported by the Nazarbayev University Faculty Development 452

Competitive Research Grant Program 2019-2021 “Kazakh Sign Language Automatic 453

Recognition System (K-SLARS)”. Award number is 110119FD4545.”

 “A.S. was awarded the funding by Nazarbayev University Faculty Development Competitive Research Grant Program 2019-2021 for the project "Kazakh Sign Language Automatic Recognition System (K-SLARS)". Award number is 110119FD4545. The funders had no role in study design, data collection and analysis, decision to publish, or preparation of the manuscript.”

Additional Editor Comments:

I am sorry that it has taken so long to reach a decision. It has been very hard to find a second reviewer for this paper. As a result, I have included the one review I received, along with my own review of the paper (below). I agree with all of R1's comments, especially the concern that the data set is not actually available for review.

- The motivation for this work is not clear. Please state more clearly the goals of: (a) automated sign language (SL) recognition; (b) automated SL translation; (c) developing corpora to train these. With regard to the last point, is the goal of developing a KRSL database to support the field of SLP generally, or more narrowly SLP related to KRSL? What kind of generalizability is desirable?

- Related to the above question, please provide a better rationale for a corpus that includes a wide range of cameras, backgrounds, and other recording details. If the purpose of a corpus is to train SLR/SLT systems to achieve high accuracy, then wouldn't more uniform recording conditions be preferable?

- There are a few questions around "native signers". It is true that, like all natural human languages, native signers should be the standard in determining what "correct" SL is, and that non-native signers are likely to produce more variable and nonstandard signing. However, it is not clear that restricting training coprora for automated SLR and SLT tasks to native signers is appropriate. This would seem to limit their generalizability, and indeed due to the variability of when people become deaf, and when, relative to becoming deaf, they learn a SL, and the historic patterns of suppression of SLs in educational systems (e.g., the residential school systems in the US and Canada), there are many more non-native than native signers in most SL communities. As a result, there is greater variability amongst signers in how signs are produced. As such, one would predict that a system trained only on native signing would generalize quite badly to much natural SL input. The question/request here is, please provide a discussion of why restricting a corpus like this to native signers is appropriate, and what the limitations are.

- p 3 contains the sentence, "Many datasets record professional interpreters who are often not native signers (i.e., CODA)". Being a CODA is not an acceptable definition of "native signer". Firstly, children of deaf adults may not be native signers because their parents don't sign. Secondly, one could be a native signer by virtue of being born to hearing parents who do sign (e.g., the child of two CODAs). Please revise this sentence, and provide an accurate definition of "native signer".

- The text of the manuscript is not clear on a very important point - the "native" status of the signers who contributed. Again, being born to a deaf parent is not a guarantee that the child is a native signer, since the parent may not sign. Thus on p. 5 the description, "They are native to KRSL since they were born and grown up in families with at least one deaf parent." is not an acceptable definition of a native signer. Please provide detail on how it was determined/confirmed that each contributor was a "native" signer.

- Related to the above, the sentences, "According to this distinction, NativeSigners-50 has 30 CODA contributors (including nine hearing signers) and 20 who are not CODA (16 deaf, one hard of hearing, and three hearing SODA). Nevertheless, we decided to name our dataset “NativeSigners-50” because all of our contributors use sign language daily, and it is their primary language of communication." are problematic. If you believe that 20/30 contributors are not native signers, then it is misleading and frankly incorrect to title your database "NativeSigners-50". It would be more appropriate to remove "native" from the title or use a numeral that accurately reflects the number of native signers who contributed.

- It seems problematic that you apparently created signs in a written language, then translated them into KRSL. SLs have their own grammars and so translations from spoken languages can result in constructions that would not be produced by native signers. Please clarify how this was addressed.

- Text makes statements that are unsupported by references. For example, p 3 states, "Many datasets record professional interpreters who are often not native signers (i.e., CODA)." But fails to provide citations to clarify what "many" is or what datasets are in question here.

- the description of approaches to SLR on p. 4 is too short, and as a result, confusing. It conflates feature extraction methods with training approaches, and reads more like a list of papers and what algorithm each used, than an explanation of the approaches, their advantages, and their limitations. This section should be revised to be more clear and detailed.

- The scope and significance of the sentence on p. 4, "All the evaluations were performed on the RWTH-PHOENIX-Weather 2014 [5] and RWTH-PHOENIX-Weather 2014T [15] datasets.", is unclear. What do you mean by "all the evaluations"? Do you mean all of the papers that you cited in this section, or only the Koller et al reference. And, what is the point of stating this? Are you implying that these data sets are considered a benchmark in the field?

- Related to the previous point, please provide a rationale for using the RWTH-PHOENIX-Weather 2014 data set as a benchmark

- Another limitation of the SLR and SLT sections on pp. 4-5 is that they do not provide the reader with any information on what benchmark levels of performance are, or what is considered "state of the art" in this field. As a result, when you describe your own methods, it is completely unclear why you chose the Stochastic CLSR and TSPNet approaches - indeed, you didn't even introduce these approaches in the "related work" section so the reader has no context as to what these are, why they were chosen, or how they compare to other approaches.

- Typically in ML, one defines the proportion of items assigned to the training, validation, and test sets a priori as percentages of the total items (e.g., 80-10-10). In the present case, (Splits 1 and 2) it does appear that your test set comprises 10% of your samples (although this is never expressed as a percentage); however your validation comprises only 4% of the data. Please provide a rationale for this low number.

- Please clarify how the RWTH-PHOENIX-Weather 2014[T] data set was used. Did you divide it into training, test, and validation sets? If so, how and in what proportions?

- I agree with R1's concerns regarding using a single split of each type. This raises significant concerns about generalizability, because your results in the present case are entirely dependent on the choice of what items were assigned to which set. By using a number of different folds you would increase generalizability, as well as the reader's confidence that your results were robust.

- Please provide more details on your methods. For instance, it is insufficent to say that "Before passing through the model, each raw image is preprocessed." What preprocessing steps were performed?

- please provide a citation for the "end2end architecture"

- please ensure that all acronyms and abbreviations are defined, and do so the first time they are used (e.g., WER, CTC)

- I am unclear on the reason for discussing the fact that "n-gram BLEU scores rapidly increase as n decreases from 4 to 1". This seems intuitive, since any training set will naturally include fewer repetitions of any 4-gram than 3-gram, etc. - and as noted, this is a property of the results on the RWTH-PHOENIX-Weather 2014T data set as well. Thus, your point seems irrelevant to your paper or your dataset (although it doesn't drop as dramatically in Splits 1 or 2). Perhaps your point has to do with the ability to generalize to signs/phrases not seen in the training set, however if so this needs to be discussed more clearly and explicitly.

- I also disagree with the statement that, "the results of TSPNet on RWTH-PHOENIX-Weather 2014T demonstrate similar behavior of rapid increase (13.41 increasing to 16.88 to 23.12 to 36.10) but with a much better BLEU-4 score (13.41 vs. 3.08)." - because you are selectively reporting only the NativeSigners-50 result for the poorest-performing split; the performance on Split 1 is actually higher than for RWTH-PHOENIX-Weather 2014T.

- Please provide some discussion of how the SLR and SLT results compare between your data set and the RWTH-PHOENIX-Weather 2014[T] one. You present comparisons of these in tables, but the significance of these is never discussed. Given that you refer to "potential limitations" of RWTH-PHOENIX-Weather 2014[T], how do you reconcile the fact that SLR/SLT performance was in general very similar between the two data sets? What would the benefit be to researchers of employing your suggestion that they report results of training applied to both your and this other data set?

- please correct the text on lines 126-127 of p. 4, which appears to be a mistaken pasting of text, or some sort of formatted article generation error.

Reviewers' comments:

Reviewer's Responses to Questions

**Comments to the Author**

1. Is the manuscript technically sound, and do the data support the conclusions?

Reviewer #1: Yes

2. Has the statistical analysis been performed appropriately and rigorously? 

Reviewer #1: Yes

3. Have the authors made all data underlying the findings in their manuscript fully available?

Reviewer #1: No

4. Is the manuscript presented in an intelligible fashion and written in standard English?

Reviewer #1: Yes

5. Review Comments to the Author

Reviewer #1: The paper presents a new dataset of Kazakh-Russian Sign Language from 50 native signers. The dataset contains native signers, different ages, different devices. Hence, it is challenging.

However, the dataset is not available. The link provided is empty. I would like to inspect the dataset, the annotations, etc. The annotation protocol is not provided.

The dataset has 3 splits: Signer independence, age, and unseen sentences. Solit 1 has 2 validation subject, 5 test subject and the rest is training. Is it just one fold or several folds? If so, how many. 1 fold is not enough because the selected 5 subjects may be too easy or too hard. Ideally, one must have, say, 10 folds, random 5 subjects in test each time. Then, one can report mean performance figures and variance.

The paper is nicely written. However, figures could be better quality with informative captions. It would be good to have successful and failing examples in split 3 as a figure. The term "signed speech" should be replaced with "sign language". You have used it once in page 1 and then at the end of page 5. Signed speech is signing each word in the order of spoken language, simultaneously with spoken language. This dataset is sign language, not signed speech or signed language.

The paper uses the RWTH-PHOENIX-Weather-2014T dataset as a sign language recogntion and translation benchmark.

Stochastic CSLR is used as a benchmark for continuous SLR. 26.1 WER is obtained on Phoenix.

TSPNet is used as a segmentation benchmark for sign language translation using multi scale attention. 13.41 Bleu4 score is obtained.

The results for the RWTH-PHOENIX-Weather-2014T dataset are the same numerical values from the respective papers. Did you print the numbers from the authors' papers directly or did you train your methods on these methods and confirm these numbers?

Camgoz et al. (Sign Language Transformers: Joint End-to-end Sign Language Recognition and Translation) obtains 24.5 WER and 21.8 BLEU4 score on the same 2014T dataset in 2020. You already have the reference in your literature section, but omitted the results.

Why does split 3 obtain better BLEU1-3 scores than split 1? WER scores from the first method do not correlate with these results. Is signer dependence the cause of this? If so would removing sentences from the same user make split 3 more realistic?

Some minor erros to be corrected:

-recognizing a real-life signed speech  recognizing real-life sign language

-were born and grown up  were born and grew up

- 173 translations  173 sentences

- too many uses of "etc."

6. PLOS authors have the option to publish the peer review history of their article (what does this mean?). If published, this will include your full peer review and any attached files.

Reviewer #1: No

---

## [Author Response · Author response to Decision Letter 0]

1 Mar 2022

Dear editor Aaron Jon Newman and the reviewer,

We are grateful for your time and reviews. We carefully addressed all the raised points and highlighted all the implemented changes with blue color in the marked up copy of the manuscript. We respond to questions and comments in the Response letter attached with this submission.

---

## [Decision Letter · Decision Letter 1]

8 Jun 2022

PONE-D-21-21778R1FluentSigners-50: a signer independent benchmark dataset for Sign Language ProcessingPLOS ONE

Dear Dr. Sandygulova,

Thank you for submitting your manuscript to PLOS ONE. After careful consideration, we feel that it has merit but does not fully meet PLOS ONE’s publication criteria as it currently stands. Therefore, we invite you to submit a revised version of the manuscript that addresses the points raised during the review process.

I have included reviews from one previous and one new reviewer. Please address their comments. Although R3 suggested "major revisions", I feel their points can be addressed with minor revisions. I think some additional context for the data set and its possible applicaitons will strengthen the paper, however I do not think this necessitates much in the way of added text. I do agree that some additional figures would enhance the paper as well.

We look forward to receiving your revised manuscript.

Kind regards,

Aaron Jon Newman

Academic Editor

PLOS ONE

Journal Requirements:

Reviewers' comments:

Reviewer's Responses to Questions

**Comments to the Author**

1. If the authors have adequately addressed your comments raised in a previous round of review and you feel that this manuscript is now acceptable for publication, you may indicate that here to bypass the “Comments to the Author” section, enter your conflict of interest statement in the “Confidential to Editor” section, and submit your "Accept" recommendation.

Reviewer #2: All comments have been addressed

Reviewer #3: (No Response)

2. Is the manuscript technically sound, and do the data support the conclusions?

Reviewer #2: Yes

Reviewer #3: Partly

3. Has the statistical analysis been performed appropriately and rigorously? 

Reviewer #2: I Don't Know

Reviewer #3: N/A

4. Have the authors made all data underlying the findings in their manuscript fully available?

Reviewer #2: No

Reviewer #3: Yes

5. Is the manuscript presented in an intelligible fashion and written in standard English?

Reviewer #2: Yes

Reviewer #3: Yes

6. Review Comments to the Author

Reviewer #2: 1. in Paper [1] did they mentioned 300 distinct sign languages?

2. Comparison with RWTH-PHOENIX-Weather database is not convincing?

3. Pasper formatting needs attention.

Reviewer #3: The paper presents a dataset of Russian sign language. This is a valuable resource as we need to recognize, document, and study different sign languages around the world. The paper needs major revisions before it’s ready for publication.

It needs clear arguments and explanations for how it can be used. What are the kinds of questions that people can answer with this dataset? Is this going to be good for linguistics and cognitive science studies for coreference resolution for example? Is the quality of the facial gesture good enough for sign simultaneously modeling? Is it only good for certain computer vision modeling tasks?

The paper needs lots of screenshots, figures, and examples so the reader can learn about the context, use-case, environment of the datacollection, etc. In its current form, it’s really hard for readers to understand the context and details of the dataset.

The anonymity issue is not discussed. If the dataset is going to be bale publicly, how are anatomizing it? What kinds of consent forms do researchers need to sign in order for them to be ale to work with the facial gestures?

How many different Russian sign language dialects exist? How diverse is your dataset? What are the limitations?

7. PLOS authors have the option to publish the peer review history of their article (what does this mean?). If published, this will include your full peer review and any attached files.

Reviewer #2: No

Reviewer #3: No

---

## [Author Response · Author response to Decision Letter 1]

11 Jul 2022

Dear editor and the reviewers,

We are grateful for your time and reviews. Below we address the points raised by the reviewers and describe the changes we implemented. All the implemented changes are highlighted with blue text color in the marked up copy of the manuscript. 

We address each comment in details below. 

Reviewer #2: 

1. in Paper [1] did they mentioned 300 distinct sign languages?

Thank you for pointing this out. Yes, they analyzed 300 works on sign language recognition, not 300 sign languages. We corrected this text in the paper. 

2. Comparison with RWTH-PHOENIX-Weather database is not convincing?

According to [1], RWTH-PHOENIX-Weather dataset is referred to as “the only resource for large vocabulary continuous sign language recognition benchmarking world wide”, although it has its own limitations. As the goal of our work is to provide an alternative dataset that could be used for sign language recognition tasks, a comparison to RWTH-PHOENIX-Weather is required. This way, the researchers can test their models’ performance on unseen signers and new sign languages.

3. Paper formatting needs attention.

Thank you. We tried to address formatting issues. 

Reviewer #3: 

The paper presents a dataset of Russian sign language. This is a valuable resource as we need to recognize, document, and study different sign languages around the world. The paper needs major revisions before it’s ready for publication.

It needs clear arguments and explanations for how it can be used. What are the kinds of questions that people can answer with this dataset? Is this going to be good for linguistics and cognitive science studies for coreference resolution for example? Is the quality of the facial gesture good enough for sign simultaneously modeling? Is it only good for certain computer vision modeling tasks?

The main purpose of this dataset is to be used as a benchmark for sign language recognition/translation architectures. It can help researchers find out if their proposed models perform well and can generalize on unseen signers and different sign languages. Additionally, the dataset can be of interest to sign language linguistics as it has real-life, linguistic and inter-signer variability. Sentence types include statements, polar questions, wh-questions, and requests. This can allow linguists to analyze the data regarding its sentence type or non-manual features. Although the dataset has its own shortcomings, we believe that it is necessary to provide alternative datasets for ML purposes to be used by the community working on sign language recognition and translation tasks. We added these details to the paper. 

The paper needs lots of screenshots, figures, and examples so the reader can learn about the context, use-case, environment of the data collection, etc. In its current form, it’s really hard for readers to understand the context and details of the dataset.

Thank you for your suggestion. We have updated the figures and added more visual examples from the dataset.

The anonymity issue is not discussed. If the dataset is going to be bale publicly, how are anatomizing it? What kinds of consent forms do researchers need to sign in order for them to be ale to work with the facial gestures?

The contributors agreed and signed consent forms for the dataset to be released publicly. It is freely available to be used for research and development purposes. 

How many different Russian sign language dialects exist? How diverse is your dataset? What are the limitations?

Russian Sign Language (RSL) is the signed language used in Russian Federation; it has dialectal variation even though it is understudied at the moment (see e.g. https://www.frontiersin.org/articles/10.3389/fpsyg.2021.740734/full ). The signed language used in Kazakhstan, which is under investigation in the current study, is, as discussed, very close to RSL lexically and most likely grammatically, which is why we use the name KRSL for it. Some other ex-Soviet countries also have sign languages that were heavily influenced by or closely related to RSL, but this is not relevant for the analysis of KRSL, so we do not discuss it. Finally, the issue of dialectal variation of KRSL in Kazakhstan has not been studied at all. As mentioned in the paper, the signers in the dataset come from different regions of Kazakhstan, so some regional variation is possibly represented in the dataset. However, it was not created for the purposes of studying dialectal variation. We added this explanation to the paper. Thank you for your question.

---

## [Decision Letter · Decision Letter 2]

15 Aug 2022

FluentSigners-50: a signer independent benchmark dataset for sign language processing

PONE-D-21-21778R2

Dear Dr. Sandygulova,

We’re pleased to inform you that your manuscript has been judged scientifically suitable for publication and will be formally accepted for publication once it meets all outstanding technical requirements.

Kind regards,

Aaron Jon Newman

Academic Editor

PLOS ONE

Additional Editor Comments (optional):

Thank you for your revisions. Although R2 made reference to unclear figures and captions in their most recent review, I believe they may not have downloaded the high-resolution TIF versions, because I find those to be quite clear and readable. Therefore, I am pleased to recommend your manuscript for publication.

Reviewers' comments:

Reviewer's Responses to Questions

**Comments to the Author**

1. If the authors have adequately addressed your comments raised in a previous round of review and you feel that this manuscript is now acceptable for publication, you may indicate that here to bypass the “Comments to the Author” section, enter your conflict of interest statement in the “Confidential to Editor” section, and submit your "Accept" recommendation.

Reviewer #2: All comments have been addressed

2. Is the manuscript technically sound, and do the data support the conclusions?

Reviewer #2: Yes

3. Has the statistical analysis been performed appropriately and rigorously? 

Reviewer #2: Yes

4. Have the authors made all data underlying the findings in their manuscript fully available?

Reviewer #2: Yes

5. Is the manuscript presented in an intelligible fashion and written in standard English?

Reviewer #2: Yes

6. Review Comments to the Author

Reviewer #2: I am satisfied with the revision but the legends of the figures are still not clear/readable. They needs to clear so that can be easily readable.

7. PLOS authors have the option to publish the peer review history of their article (what does this mean?). If published, this will include your full peer review and any attached files.

Reviewer #2: No

---

## [Editor Report · Acceptance letter]

2 Sep 2022

PONE-D-21-21778R2 

FluentSigners-50: a signer independent benchmark dataset for sign language processing 

Dear Dr. Sandygulova:

I'm pleased to inform you that your manuscript has been deemed suitable for publication in PLOS ONE. Congratulations! Your manuscript is now with our production department. 

Kind regards, 

on behalf of

Dr. Aaron Jon Newman 

Academic Editor

PLOS ONE